# Impact of Different Metals on the Performance of Slab Tamm Plasmon Resonators

**DOI:** 10.3390/s20236804

**Published:** 2020-11-28

**Authors:** Gerald Pühringer, Cristina Consani, Bernhard Jakoby

**Affiliations:** 1Institute for Microelectronics and Microsensors, Johannes Kepler University, 4040 Linz, Austria; bernhard.jakoby@jku.at; 2Silicon Austria Labs GmbH, 9524 Villach, Austria; Cristina.Consani@silicon-austria.com

**Keywords:** thermal emitter, tamm plasmons, silicon photonics, mid-infrared

## Abstract

We investigate the concept of slab Tamm plasmons (STP) in regard to their properties as resonant absorber or emitter structures in the mid-infrared spectral region. In particular, we compare the selective absorption characteristics resulting from different choices of absorbing material, namely Ag, W, Mo or highly doped Si. We devised a simplified optimization procedure using finite element simulations for the calculation of the absorption together with the application of micro-genetic algorithm (GA) optimization. As characteristic for plasmonic structures, the specific choice of the metallic absorber material strongly determines the achievable quality factor (*Q*). We show that STP absorbers are able to mitigate the degradation of *Q* for less reflective metals or even non-metals such as doped silicon as plasmonic absorber material. Moreover, our results strongly indicate that the maximum achievable plasmon-enhanced absorption does not depend on the choice of the plasmonic material presuming an optimized configuration is obtained via the GA process. As a result, absorptances in the order of 50–80% could be achieved for any absorber material depending on the slab thickness (up to 1.1 µm) and a target resonance wavelength of 4.26 µm (CO_2_ absorption line). The proposed structures are compatible with modern semiconductor mass fabrication processes. At the same time, the optimization procedure allows us to choose the best plasmonic material for the corresponding application of the STP structure. Therefore, we believe that our results represent crucial advances towards corresponding integrated resonant absorber and thermal emitter components.

## 1. Introduction

Absorption sensing of fluids or gases via highly integrated on-chip devices is a field of high interest for the industry. Successfully reducing the detection limit remains the critical issue for such small devices. Highly efficient components (emitter, coupler, waveguides) of an on-chip absorption sensor are required for this task, while compatibility with mass fabrication techniques should be maintained. Various approaches for resonant or non-resonant absorber or emitter designs are an intensely investigated field of research and include structured metallic gratings, photonic crystals or layered structures [1,2,3,4,5,6,7]. The mid-infrared (IR) region offers many applications regarding optical absorption sensing and is greatly compatible with the use of silicon-based material, as Si compounds feature very high transmittance in this region. In addition to their fabrication compatibility with standard cleanroom processing, these concepts are required to be integrable into on-chip sensing components, which may include, e.g., coupling to waveguides. The gold standard in terms of bandwidth, radiation density and integrability is quantum-cascade lasers (QCLs) [8,9]. However, the associated fabrication costs per device are rather high, which makes them unappealing for mass fabrication. As a result, approaches utilizing thermal radiation, such as Tamm plasmon-polariton (TP) emitters featuring extended layers with alternating thickness, have been gaining a lot of attention recently [10,11,12]. Despite the very feasible fabrication of TP structures, the layered design is very difficult to integrate into an on-chip sensor system, as efficient coupling to a waveguide is required. In order to address this issue, we recently proposed a concept utilizing TP structures in a slab scenario, so called slab Tamm plasmon (STP) structures [13], as illustrated in Figure 1a,b. They feature good compatibility with standard complementary metal-oxide-semiconductor (CMOS) fabrication processes and, at the same time, facilitate inherent coupling to a dielectric slab waveguide. Etching laterally exactly defined areas into the silicon slab creates regions with alternating indexes of refraction forming a Bragg mirror or, alternatively, an aperiodic bandgap structure [13,14]. In order to keep the analogy to conventional 1D-TP layers, we refer to these regions as layers. The silicon slab is suspended in air via a membrane formed by Si nitride, as commonly employed previously; see, e.g., [15] or [16]. The resonator structure terminates at a metal (or heavily doped Si) serving as the reflective and absorbing element (mirror material). Thus, the basic composition and fundamental working principle of STP structures is analogous to the conventional one-dimensional TP structure ([3,17]). In contrast to TP resonances, STP resonances only support transverse electric (TE) polarization (i.e., out-of-plane electric field; see arrows in Figure 1a). Therefore, the plasmonic absorption (and emission) is expected to be highly polarized [13,18]. By Kirchhoff’s law, these structures can be used as either emitters or detectors [3,13,19]. The integration of such a structure is feasible via modern CMOS processes (e.g., lithography together with “lift-off” and “dry-etching” processes), particularly because of the arbitrary extent of the metal layer in the z-direction (”Distributed Bragg Reflector side” emission/absorption). We believe that STP structures have the potential to be a low-cost alternative to expensive quantum cascade lasers (or quantum-well photodetectors), as integrated waveguide sources or detectors in the mid-infrared region.

In this paper, we show that the concept of STP structures aligns well with the usage of less reflective metals or heavily doped silicon. Typically, more reflective metals like silver or gold enable high-*Q* resonances, not only in the case of TP structures [1,20,21,22]. Ag and Mo are highly reflective metals in the mid-IR region, enabling high *Q* factors, whereas W is a commonly used as a “lower *Q*” plasmonic material featuring excellent mechanical and thermal properties in return [5,14,23]. Heavily doped silicon is also widely investigated in the context of plasmonic application and presents the extreme case of a less reflective absorbing mirror material for STP structures. We focus on Ag, W, Mo and doped Si as mirror materials of the STP structures in order to cover a wide range of different plasmonic properties. Mo and W seem like very promising candidates for real-world applications due to their low thermal expansion coefficients and high melting points compared to Ag or Au, and these factors are crucial for the stability of the membrane at higher temperatures, for example.

In the case of conventional (1D) TP structures, by fulfilling the critical coupling condition (i.e., matching the number of layers to the corresponding mirror material), one is always able to achieve unity absorption at the resonance wavelength independent of the metal choice. In contrast, unity absorptance is never achievable for STP structures (and slab resonators in general) due to radiation losses (see, e.g., [24,25]). This raises the question: does a lower *Q* factor also lead to a lower plasmon enhanced absorption when the structure is optimized for a given mirror material and slab height? If yes, this would also limit the maximum achievable power in the case of an emitter application. This work will show that this is not the case and that a feasible resonator structure can be achieved even for heavily doped silicon instead of a metal, in principle. Together with the consideration that the shortcomings of the fabrication processes in terms of accuracy exacerbate a high yield of well-performing devices, designs with lower *Q* resonances may be the more reliable option.

The design for a structure including heavily doped Si as mirror required a small change in the design, as it is technically not possible to create a sharp undoped–doped interface (due to the diffusion of dopants). Thus, a small airgap with width $d_{5}$ next to the doped Si was added (see Figure 1b).

In this paper, we focus on the target resonant absorption wavelengths $\lambda_{0}$ = 4.26 µm together with slab thicknesses *t* from 0.7 to 1.1 µm for each wavelength. The parameters provide almost mono-mode characteristics, a low fraction of the electric field outside the silicon and, thus, long propagation lengths for the corresponding slab waveguides. The corresponding fundamental guided slab modes feature confinement factors (a measure of device sensitivity; see [26,27]) of 2.2%, 1.2% and 0.8% for *t* = 0.7, 0.9 and 1.1 µm, respectively. This is an order of magnitude which has been proven to be suitable for integrated absorption ${\mathrm{CO}}_{2}$ sensing, for example [16,28]. At the same time, these slab thicknesses provide sufficient light confinement for STP resonances and are feasible for CMOS fabrication processes.

This work is structured as follows: first, we study the dielectric properties of Ag, W, Mo and heavily doped Si. Then, we find performant starting configurations before employing genetic algorithm optimization in order to find the best possible dimensions for different sets of given parameters per STP structure (i.e., slab thickness and mirror material), respecting the constraints of the fabrication process. Moreover, we discuss the impact of the membrane configuration (i.e., membrane thickness and index contrast) on the resonant mode. We analyze the differences between the particular mirror materials in terms of bandwidth and maximum absorptance and deduce recommendations for device fabrication. Further, we discuss the effect of increasing the material operation temperature by applying temperature models of the dispersion relation.

## 2. Materials and Methods

### 2.1. Mirror Materials

The optical dispersions of the materials at room temperature were obtained from well-established data from the literature including intrinsic/doped Si, SiO_2_, Si_3_N_4_, W, Mo and Ag [29,30,31,32,33,34,35]. Among these, the metals as well as the doped Si were used as mirror materials for the STP structures. Some studies suggested heavily doped Si (together with other semiconductors) as suitable plasmonic material for the mid- to far-infrared region, particularly in the context of surface plasmon resonances (SPR) and localized surface plasmon resonances (LSPR) [36,37,38]. Our previous study with doped Si in STP structures successfully demonstrated the feasibility of this concept [39]. The dispersion models allow us to estimate the level of suitability of each material for plasmonic applications. A useful measure for the maximum quality factor of a certain metal in plasmonic applications is the reflectance $R_{M}$. Subsequently, plasmonic devices employing silver or gold feature the highest *Q* factors possible, whereas devices with less reflective metals or alternative plasmonic materials never can reach the same levels of field enhancement [20]. This also applies for the mirror materials of TP resonator structures [3,4,12]. In Figure 2a,b, the reflectance for an air–metal interface $R_{M}\left(n,k\right)=\left({\left(n\;-1\right)}^{2}+k^{2}\;\right)\;/\left({\left(n\;+\;1\right)}^{2}+k^{2}\right)$ and the figure of merit for the achievable *Q* factor proposed in [5].
(1)$${\mathrm{FOM}}_{Q}=\frac{1}{A_{M}}=\frac{1}{1-R_{M}}=\frac{{\left(n+1\right)}^{2}+k^{2}}{4n},$$
based on the absorptance of the metal $A_{M}$ and refractive indices modeled in [33] (n is the real and k is the imaginary part, respectively) are plotted. The expression $\frac{k^{2}}{n}$ used in [5] is only valid if $k^{2}\gg n^{2}$ holds, which is not true in case of doped semiconductors. It can be seen that the values for ${\mathrm{FOM}}_{Q}$ and $R_{M}$ are substantially lower for the heavily doped Si compared to the metals. The doping level has a major impact on the suitability for plasmonic applications for a target resonance wavelength. The highest doping levels achievable (after annealing) are in the range of ~$2.8\times 10^{20}$ cm^−3^ (phosphor-doped), which enables plasmonic applications in the mid/far-infrared region (7–14 µm) [40]. However, for a target resonant wavelength $\lambda_{0}=4.26$ µm (see red vertical line in Figure 2a), this doping concentration is insufficient for plasmonic applications, as reflected by the green dashed line in Figure 2a,b (intersection with red vertical line). Although doping levels beyond this value are yet to be achieved (there are several approaches currently under investigation, e.g., [41]), we used the value of $10^{21}$ cm^−3^ in the model provided by [33] for the sake of comparability, as this doping level is just enough to enable plasmonic resonances without changing the target resonance wavelength or t. Despite the high doping level, real metals feature values of ${\mathrm{FOM}}_{Q}$ around one order of magnitude higher (at 300 K) together with values for $R_{M}$ larger than 0.95. The authors of [20,37,42] highlight the limits of less reflective metals or doped semiconductors (Si) in terms of the inevitable intrinsic energy losses of the free carriers, which is a circumstance that is exploited by TP absorber or emitter structures. Switching off all intrinsic losses would lead to zero field enhancement (no resonance), as all incident power would be radiated away or be reflected. Particularly for STP structures, the fulfilment of the STP resonance condition is facilitated by increased ohmic losses, as the intrinsic plasmonic loss of the metal has to be the dominating damping mechanism of the STP resonance in relation to the damping by radiation [13,17]. It has to be emphasized that this does not imply higher plasmon-enhanced absorption at resonance for less reflective mirror materials, as also Ag provides enough loss for realizing optimized STP dimensions, which enables a compensation of the lower loss via a higher field enhancement. Moreover, it is worth noting that the lower *Q* resonances are less susceptible to deficiencies of the fabrication process, which includes the accuracy of the lithographic process, etching angle or material defects. This is particularly important as perfect edges with no surface roughness are considered in the simulations.

### 2.2. Simulation Domain and Optimization

Optimization of the STP structures is achieved by variation of the individual widths of the Si/Air regions $d_{1}$ to $d_{4}$ formed by etching of the slab. An initial configuration is created by modifying a Bragg reflector structure, which enables initialization of the genetic algorithm optimization process, resulting in an aperiodic configuration thereafter.

#### 2.2.1. Initialization of Modified Quarter Wave Stack

The structure was set up as an interface of a quarter wave stack (QWS) and the plasmonic mirror material. A TE polarized guided slab mode was launched from the left boundary onto the STP stack featuring layer thicknesses *d*_*i*_ (*i* = 1…4) and the mirror material (i.e., no transmission through the latter). A total of four dielectric layers were chosen as degrees of freedom for the optimization process. In the case of a conventional TP structure, the optimal number of layers is determined by the reflectance of the mirror material (usually a metal) and the refractive index contrast of the layer [3]. In the case of an STP structure, t together with $\lambda_{0}$ impacts the optimal number additionally. For ratios $\frac{\lambda_{0}}{n_{\mathrm{Si}}w}$ near, yet slightly above, unity, a count of four layers has proven to provide strong STP resonances for Ag at elevated temperatures [13]. It has to be noted that this rather low number of layers may not be optimal for very highly reflective metals like Ag at room temperature.

As one-dimensional TP structures are usually composed of quarter wave stacks, the initial configuration can be, in principle, set up with $d_{1}=d_{3}=\frac{\lambda_{0}}{4}$ (air layers) and $d_{2}=d_{4}=\frac{\lambda_{0}}{4n_{\mathrm{eff}}}$ (Si layers; $n_{\mathrm{eff}}$ is the effective refractive index of the guided slab mode). However, these values do not work well with STP structures, as the situation is fundamentally different for the air layers $d_{1}$ and $d_{2}$ compared to a conventional TP structure featuring plane-waves: there is no meaningful way to define $n_{\mathrm{eff}}$ for the air layers, as the field of the STP mode can neither be assigned to a guided mode nor to a plane-wave. Moreover, the small spatial extent of $d_{4}$ is not able to confine the STP interface state sufficiently, which leads to strong coupling with free space radiation. Therefore, a simple quarter wave stack does not lead to a sufficient fulfilment of the resonance condition for STP structures (see resonance condition for TP structures in, e.g., [17]).

Table 1 shows the altered initial layer thicknesses from a conventional QWS to a quasicrystal-like “modified QWS”, where $\delta_{p}$ denotes the penetration depth and $n_{\mathrm{Si}}$ and $n_{\mathrm{MM}}$ denote the refractive indices of silicon and mirror material, respectively. The penetration depth $\delta_{p}$ can be calculated via the material dispersion as
(2)$$\delta_{p}=\frac{\lambda_{0}}{4\pi \;k_{\mathrm{MM}}}$$
where $k_{\mathrm{MM}}$ is the imaginary part of the complex refractive index (N=n+ik) of the mirror material. The specific values can be explained as follows: the effective cavity composed of $d_{4}$ was tripled in order to enhance light confinement by increasing the spatial extent of the plasmonic resonance. Moreover, the values for the air gaps *d*_1_ and *d*_3_ were multiplied by 0.5 in order to reduce the coupling with radiation modes and keep the optical path nearly constant. Thus, the two air gaps $d_{1}$ and $d_{3}$ induce a phase shift of π/4 each, corresponding to a total phase shift of a quarter wavelength layer of π/2. This semi-empiric procedure yields a strong resonance at the desired wavelength $\lambda_{0}$ depending on the slab thickness t, as well as on the choice of mirror material. Despite using $n_{\mathrm{eff}}$ for the definition, a small scaling factor had to be applied additionally in order to tune the resonance wavelength to the value of 4.26 µm due to the hybrid localization of the STP mode in two directions. Figure 3a demonstrates the strong spectral response of the initial configuration for a 900 nm Si slab and Mo. Interestingly, the refined configuration features similar characteristics when calculated with purely one-dimensional layers, as shown by the orange solid lines in Figure 3a. Notably, the maximum absorptance is very similar in both scenarios (slab and 1D), while the bandwidth is narrower for the STP scenario. The narrowing is due to the altered shape of the STP mode profile compared to the mode profile in the plane-wave scenario, as already discussed in [13]. The very similar performance in terms of plasmonic field enhancement and absorptance at resonance $a\left(\lambda_{0}\right)$ reflects the suppressed impact of radiation loss in the STP scenario. While it is easy to enhance $a\left(\lambda_{0}\right)$ to near unity for a conventional 1D TP structure by, e.g., adjusting the number of QW layers, this structure does not provide resonance in the STP scenario.

#### 2.2.2. Genetic Algorithm Optimization

The configuration obtained from the procedure described above was used as the initial configuration for the GA optimization in the slab scenario utilizing finite element (FE) software (COMSOL Multiphysics). In analogy to [13], the net power flux between the dielectric and the mirror material was calculated via Poynting’s theorem (in frequency domain) at $\lambda_{0}$. The corresponding fitness function does not feature explicit improvement of the *Q* factor (i.e., narrowing of the full-width-at-half maximum $\Delta \lambda_{\mathrm{fwhm}}$). However, optimization of the resonant plasmon-enhanced absorptance is more important in the case of STP structures, as this value is crucial for the power yield or the sensitivity for real-world application of the STP structure. Due to the altered mode properties compared to conventional TP modes, STP modes feature excellent values for the *Q* factor inherently. Figure 3b (red solid lines) shows the field profile as well as the spectral response of the GA optimized configuration (in the inset) for a Si slab of 900 nm height and Mo. Compared to the initial configuration (refined QWS, red solid line Figure 3a), the field profile is very similar, yet the maximum absorptance could be improved from 0.65 to 0.7 (note the logarithmic scale in the inset). This improvement is reflected by the slightly stronger field enhancement next to the Si–Mo interface. Thus, the constraint GA procedure does not change the resonant STP mode but refines the impedance match for the initial STP mode defined by the modified QWS. The variation of the layer thicknesses was constrained for the GA optimization process. The upper $d_{i}^{+}$ and lower boundaries $d_{i}^{-}$ were set as $d_{i}^{\pm }\;=d_{i}^{\mathrm{INI}}\left(1\pm 0.5\right)$ for the initial layer thickness $d_{i}^{\mathrm{INI}}$ of the *i*-th layer, where *i* = 1…4. This guarantees fast convergence to a strong global minimum for the fitness function. The magnitude of the improvement through the GA optimization strongly depends on t and the reflectance of the mirror material. The slower computation speed of the FE frequency domain solver (compared to the Transfer Matrix Method) demanded so-called micro-genetic algorithm (µ-GA) optimization featuring a population of only eight individuals [43]. The specific parameters’ operations for the µ-GA optimization process (tournament selection and two-point crossover) are unchanged from [13]. The optimization was performed for each individual mirror material and slab thickness t, as the optimal values of $d_{i}$ (i=1…5) depend on each individual configuration.

## 3. Results

### 3.1. Silver

The results of the spectral response for the optimized configurations with Ag as mirror material are summarized in Table 2 and Figure 4. The maximum absorptance at resonance $a\left(\lambda_{0}\right)$ ranges from ~0.52 to 0.70 depending on t. These values are slightly lower compared to the configurations with the other mirror material featuring lower reflectance and ${\mathrm{FOM}}_{Q}$. In turn, the STP structures featuring silver feature the best *Q* factor, as expected from (1) and Figure 2b. The slightly lower values for $a\left(\lambda_{0}\right)$ compared to the other mirror materials are the result of only three “effective layers” in the initializing modified QWS prior to optimization, which is insufficient for optimal fulfillment of the resonance condition for Ag or Au [3,5]. As discussed in Section 2.2.2, the GA optimization is not able to fundamentally change the resonance or to compensate for the non-optimal impedance match completely. It should be noted that the optical dispersion of Ag considers the material to be at room temperature. However, at elevated temperatures, the reflectance decreases, which results in an improvement in the impedance match. Thus, in the case of operating the STP structure as a thermal emitter (and values of *t* near 1 µm), four layers are sufficient, as shown in our previous work [13].

Although silver features excellent optical properties and is rather simple to process with standard technology, elevated temperatures remain an issue due to thermal expansion (stress in membrane) and oxidation.

### 3.2. Tungsten

The lower ${\mathrm{FOM}}_{Q}$ and reflectance of W enabled significantly better impedance match and, thus, higher values for $a\left(\lambda_{0}\right)$ compared to Ag. As a result, the constraint to four layers fits better for tungsten for achieving a maximum absorptance up to 0.80, as shown in Figure 5 and Table 2. This improved value is in agreement with previous studies considering tungsten as mirror material [3,5]. Although the *Q* factors are expectedly lower compared to the Ag structures (by ~−25%), the decrease is not so pronounced as suggested in the previous references (~above 50%, depending on $\lambda_{0}$). This lowered decrease can be related to the shape of the STP mode profile, which results in a lowered electric field–plasmon interaction compared to a plane-wave scenario. This, in turn, lowers the impact of the metal on the total resonance damping [13]. The same argument can be used as explanation for the improved *Q* when comparing STP resonators to conventional (1D) TP resonators in the plane-wave scenario. The reduction in FOM_Q_ for W compared to Ag suggested by Figure 2 would result in a decrease in *Q* beyond 50%, which compares to the observed reduction of only ~25%.

It is also interesting to note the reduced relative electric field enhancement at resonance in Figure 5b,c when compared to the field amplitude for the Ag structure in Figure 4b,c. The amplitude of the relative field enhancement is mainly a direct consequence of the *Q* factor of the resonance. Although the magnitude of this amplitude is the reason for the enhancement of $a\left(\lambda_{0}\right)$, it seems that there is not a strong correlation between $a\left(\lambda_{0}\right)$ and the field amplitude. This can be explained via a combination of three effects determining $a\left(\lambda_{0}\right)$: (I) the coupling efficiency to the (S)TP mode (resonance condition; see [13,24]), (II) the penetration depth of the field into the metal (2) and (III) the presence of radiation losses. The *Q* factor is mainly determined by effect (II) only, just as for conventional TP structures. These effects are discussed in more detail in Section 4.

Tungsten should be more suitable for thermal emitter applications due to the higher melting point and lower thermal expansion coefficient. The latter reduces stress on the proposed silicon–nitride membrane (see Figure 1).

### 3.3. Molybdenum

As Mo features excellent values for ${\mathrm{FOM}}_{Q}$ according to Figure 2, the results for the optimized configurations shown in Figure 6 are expectedly similar to the Ag-based structures. The high melting point of Mo facilitates the application of the STP resonator as emitter.

It has to be pointed out that the optical properties of Mo are based on the data of Ordal et al. [34], which are in agreement with Minissale et al. [44]. However, the values for the negative real part of the permittivity reported in [5] are in sharp disagreement with the previous references, which would result in much lower values for the reflectance and ${\mathrm{FOM}}_{Q}$. It can be assumed that this discrepancy is related to the characteristics of the surface profile of the Mo film, as carefully processed molybdenum mirrors are proven to show very high reflectance [45], in agreement with [34,44] for the mid-IR region. As these references specifically focus on studying the optical properties of Mo, we choose to use a Lorentz–Drude model with the optical parameters of [34].

Notably, the high values for the reflectance and ${\mathrm{FOM}}_{Q}$ of Mo, enabling resonances with *Q* factors of ~0.75, together with the excellent mechanical properties and high melting point, suggest Mo to be a promising compromise between Ag and W. For the case of a resonant absorber (where Mo is at room temperature), an additional layer may be beneficial in order to maximize the values for $a\left(\lambda_{0}\right)$. In the case of an application as a thermal emitter (where Mo is at elevated temperatures), the reflectance and ${\mathrm{FOM}}_{Q}$ decrease according to various temperature models ([44,46]) and, subsequently, an STP structure composed of four layers is sufficient for a good impedance match.

### 3.4. Heavily Doped Silicon

For the STP structure featuring heavily doped Si as mirror material, an additional airgap with width $d_{5}$ next to the Si is introduced, as illustrated in Figure 1b. This is necessary, as it is not possible to fabricate a sharp interface between doped and undoped Si due to the diffusion of dopants. The size of the effective cavity (= region of field enhancement) is set as $d_{\mathrm{eff}}=d_{4}+d_{5}$ and is now defined as variable layer thickness for the GA optimization process in place of $d_{4}$ in the case of Ag, W or Mo. With the obtained value for $d_{\mathrm{eff}}$, the specific value for the corresponding silicon and air layer is set at $d_{4}=0.95d_{\mathrm{eff}}$ and $d_{5}=0.05n_{\mathrm{Si}}d_{\mathrm{eff}}$, respectively.

The results obtained using the optical dispersion data of heavily doped Si (doping concentration ~10^21^ cm^−3^) by Basu et al. [33] are shown in Figure 7. Notably, the value for the reflectance is nearly 20% and ${\mathrm{FOM}}_{Q}$ is approximately an order of magnitude lower compared to the metals (at $\lambda_{0}=4.26$ µm). Despite the significantly impaired optical properties and the additional airgap, similar values of $a\left(\lambda_{0}\right)$ compared to the W-based STP structures of up to 0.82 (depending on t) could be obtained. In turn, the values for *Q* are expectedly much lower compared to the metal-based STP structures.

Doped silicon as mirror material would have significant advantages in STP absorber or emitter structures due to the same thermal expansion coefficient as the surroundings Si-based materials (lower stress on the membrane), extremely high melting point and chemical inertness (no oxidation). However, reaching doping levels in the range of ~$10^{21}$ cm^−3^ has yet to be accomplished, as already discussed in Section 2.

## 4. Discussion and Conclusions

While our results are in agreement with previous studies characterizing the plasmonic properties of various materials in the mid-IR region, they also show significant differences to conventional 1D TP structures. Although the inevitable radiation losses (due to leakage) prevent unity absorption, the STP mode shape results in higher *Q* factors compared to 1D TP structures (see inset of Figure 3) due to hybrid localization in two directions. As a result, the radiation loss together with the hybrid localization lowers the impact of the internal plasmonic material losses on the resonance. This results, on one hand, in a lower bandwidth ${\Delta \lambda }_{\mathrm{fwhm}}$ (see inset of Figure 3a, already discussed in [13]) and, on the other hand, in a less pronounced increase in ${\Delta \lambda }_{\mathrm{fwhm}}$ when employing a less reflective mirror material when compared to corresponding conventional 1D TP resonators. This makes STP structures very suitable for high-*Q* resonances with less reflective mirror materials. As mechanical properties like melting point, thermal expansion (stress on membrane) or oxidation of these materials (i.e., W, Mo, doped Si) are usually preferable to the properties of Au or Ag, STP structures are particularly interesting together with these robust materials.

The approach of a modified QWS has shown to provide fairly strong resonances (at least above 50% absorptance, strongly dependent on t and the mirror material) already prior to the GA optimization, as can be extracted from Table 2 via the value $a\left(\lambda_{0}\right)-\Delta a_{\mathrm{GA}}$. The improvement of the absorptance by the optimization $\Delta a_{\mathrm{GA}}$ reflects a refinement of the impedance match (extent is dependent on t). The optimization cannot fundamentally change the resonance, as the constraints limit the optimization space and the resulting small deviation from a periodic configuration does not change the electric field profile at resonance significantly (see Figure 3). At the same time, these constraints enable a reliable optimization for each type of mirror material and slab height. In particular, the µ-GA algorithm appears to be a reliable tool for finding the best configuration (within fabrication constraints), considering every parameter impacting the STP resonance. For example, a different membrane configuration impacts the power loss by radiation just as a different number of STP layers or refractive index contrast.

Our study indicates that the *Q* factor of an STP resonance (mainly determined by the mirror material) does not correlate with the maximum $a\left(\lambda_{0}\right)$ achievable for an optimized configuration. Values of ~0.65–0.80 for $a\left(\lambda_{0}\right)$ could be obtained for W and doped Si, while these values are slightly lower for the highly reflective metals Ag and Mo (~0.55–0.75). This decrease for Ag and Mo is explained by the insufficient number of four layers, which leads to a lightly suboptimal impedance match for these materials. However, employing temperature models reduces the reflectance (and ${\mathrm{FOM}}_{Q}$), which can improve the value of $a\left(\lambda_{0}\right)$ to up to ~0.8 [13].

The relative field enhancement of an STP resonance is reflected dominantly by *Q*. Changing the mirror material (or, equivalently, the ohmic losses) also changes *Q* and thus mainly determines the magnitude of the relative field enhancement. Notably, also $a\left(\lambda_{0}\right)$ is influenced by the field amplitude by changing t, as can be seen by comparing Figure 4b,c, Figure 5b,c, Figure 6b,c and Figure 7b,c. For each mirror material, the value for $a\left(\lambda_{0}\right)$ and the relative field enhancement varies with the slab thickness t while *Q* stays approximately constant. The influence of the µ-GA optimization on $a\left(\lambda_{0}\right)$ and the field enhancement can be observed in Figure 3 (note the dashed horizontal line).

The value for $a\left(\lambda_{0}\right)$ is determined by three factors, namely by (I) the fulfillment of the resonance condition, by (II) the ohmic losses of the metal and by (III) the radiation losses (for STP structures exclusively). As for conventional TP structures, the ohmic loss of the mirror material mainly determines how much field enhancement is required in order to achieve a particular value for $a\left(\lambda_{0}\right)$. Equivalently, the ohmic losses determine the penetration depth of the field into the mirror material (2). In the case of STP resonances, the presence of radiation loss limits the value of $a\left(\lambda_{0}\right)$.

Remarkably, values of $a\left(\lambda_{0}\right)$ of up to 0.8 can also be achieved for heavily doped silicon. It has to be pointed out that a target resonance wavelength of 4.26 µm requires extremely high doping levels in the order of magnitude of $10^{21}$ cm^−3^, which is slightly above the absolute limit of what could be achieved so far [33,40]. Silicon featuring this doping level can be seen as a worst-case scenario in terms of ohmic losses and *Q* of the corresponding structure for this target wavelength. Although there exist approaches to overcoming the doping limit (e.g., [41]), the results may be even more interesting for larger resonant wavelengths (7–14 µm) when scaling the STP structure accordingly. In this case, feasible doping levels (e.g., 2.8 × $10^{20}$ cm^−3^, as achieved in [33]) can be applied, enabling high reflectance and thus plasmonic applications, as studied in [36,40], for example. Notably, the main downside of low propagation lengths for non-metal plasmonic materials does not apply for STP structures (or TP structures in general), as opposed to waveguides based on surface plasmons [20].

Possible applications as a detector could be realized via integration in pyroelectric or thermopile structures [47]. Although the STP emitters couple inherently to slab-waveguide modes, additional coupling to other waveguide types (e.g., slot-waveguide array, plasmonic waveguides, etc.) might be feasible in order to overcome the detection limits of evanescent wave sensing.

Although not yet available, experimental verification regarding the compatibility with semiconductor fabrication processes and optical properties is currently under intense investigation and planned for the near future.

## Figures and Tables

**Figure 1 sensors-20-06804-f001:**
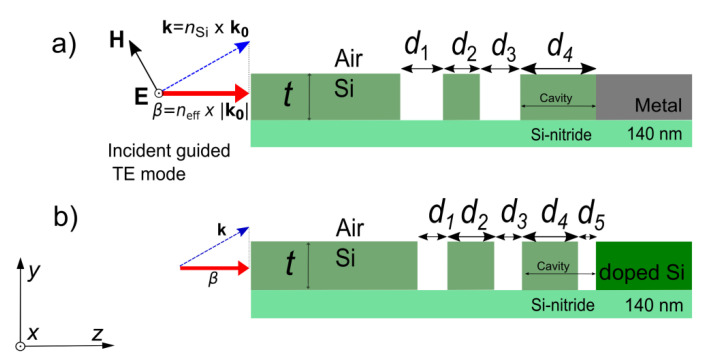
(**a**) Schematic illustration of a slab Tamm plasmon structure with slab height t featuring a metal as plasmonic material. (**b**) Corresponding slab TP structure featuring strongly doped silicon as plasmonic material and an additional airgap with width $d_{5}$. The incident guided mode with effective index $n_{\mathrm{eff}}$ is TE polarized (i.e., electric field is out-of-plane).

**Figure 2 sensors-20-06804-f002:**
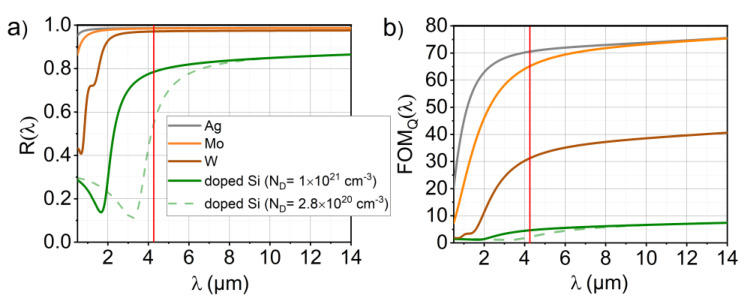
(**a**) Calculated reflectance and (**b**) figure of merit for the *Q* factor FOM_Q_ in the mid-IR spectral region for different mirror materials based on the Lorentz–Drude models from the literature. The vertical red line indicates target resonance wavelengths $\lambda_{0}=$ 4.26 µm. The doping concentration $N_{D}$ for the Si dispersion is denoted in the legend. The green dashed line shows the dispersion for the highest doping level currently achievable.

**Figure 3 sensors-20-06804-f003:**
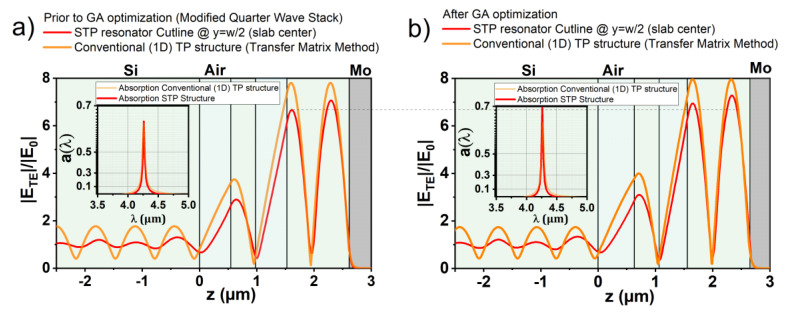
Comparison of relative field enhancement between the conventional 1D TP structure (orange solid line) and the STP structure (cross-section through the 0.9 µm Si slab center, red solid line) for the modified quarter wave stack with Mo at λ_0_. (**b**) Analogous to panel (**a**) for the optimized configuration via the GA process. The insets show the corresponding spectral responses of the structures.

**Figure 4 sensors-20-06804-f004:**
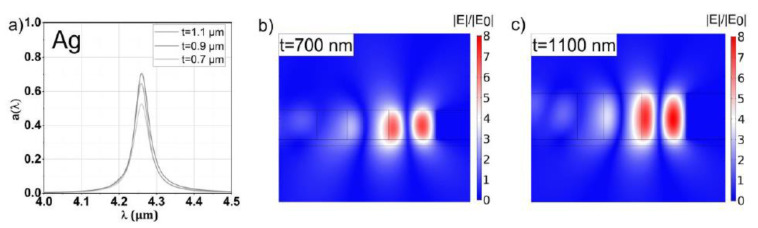
(**a**) Spectral response of the µ-GA optimized Ag-based STP structures for slab thicknesses of 0.7, 0.9 and 1.1 µm, respectively, and a resonant wavelength of 4.26 µm. The relative field enhancements of the thinnest (0.7 mm) and thickest (1.1 µm) slab are shown in panels (**b**,**c**), respectively. The high Q factor of the resonance is reflected by a 6- to 8-fold enhancement of the electric field depending on t.

**Figure 5 sensors-20-06804-f005:**
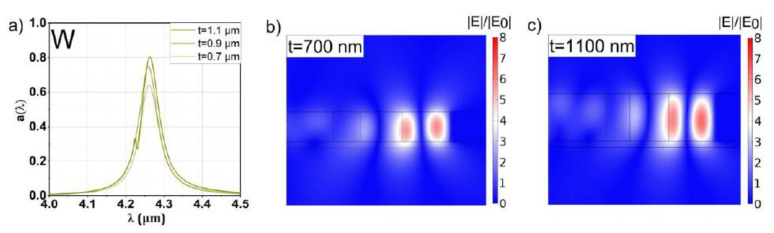
(**a**) Spectral response of the µ-GA optimized W-based STP structures for slab thicknesses of 0.7, 0.9 and 1.1 µm, respectively, and a resonant wavelength of 4.26 µm. The relative field enhancements of the thinnest (0.7 mm) and thickest (1.1 µm) slab are shown in panels (**b**,**c**), respectively. The Q factor of the resonance is reflected by a 5- to 6-fold enhancement of the electric field depending on t

**Figure 6 sensors-20-06804-f006:**
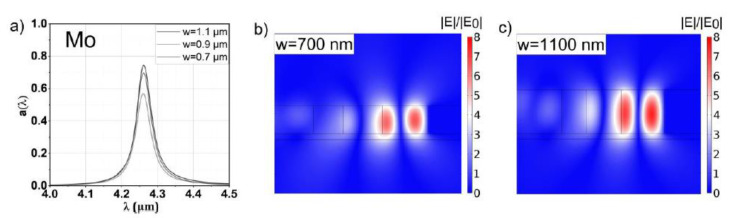
(**a**) Spectral response of the µ-GA optimized Mo-based STP structures for slab thicknesses of 0.7, 0.9 and 1.1 µm, respectively, and a resonant wavelength of 4.26 µm. The relative field enhancements of the thinnest (0.7 mm) and thickest (1.1 µm) slab are shown in panels (**b**,**c**), respectively. The high Q factor of the resonance is reflected by a 6- to 8-fold enhancement of the electric field, similar to the resonance featuring Ag.

**Figure 7 sensors-20-06804-f007:**
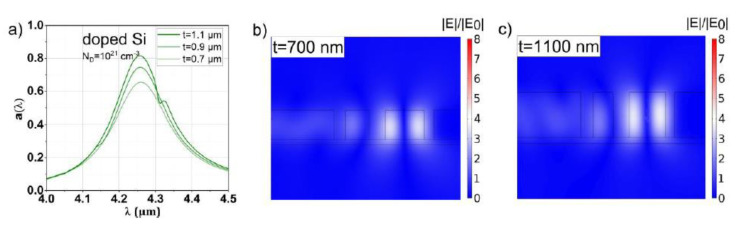
(**a**) Spectral response of the µ-GA optimized Si-based STP structures for slab thicknesses of 0.7, 0.9 and 1.1 µm, respectively, a resonant wavelength of 4.26 µm and a doping concentration of 10^21^ cm^−3^. The relative field enhancements of the thinnest (0.7 mm) and thickest (1.1 µm) slab are shown in panels (**b**,**c**), respectively. The high Q factor of the resonance is reflected by a 3- to 4-fold enhancement of the electric field similar to the resonance featuring Ag.

**Table 1 sensors-20-06804-t001:** Layer thicknesses *d*_1_–*d*_4_ for the conventional and modified quarter wave stack, respectively.

	*d* _1_	*d* _2_	*d* _3_	*d* _4_
conventional QWS	$\frac{\lambda_{0}}{4}$	$\frac{\lambda_{0}}{4n_{\mathrm{eff}}}$	$\frac{\lambda_{0}}{4}$	$\frac{\lambda_{0}}{4n_{\mathrm{eff}}}$
modified QWS	$\frac{\lambda_{0}}{8}+\delta_{p}$	$\frac{\lambda_{0}}{{4n}_{\mathrm{eff}}}+n_{\mathrm{Si}}\delta_{p}$	$\frac{\lambda_{0}}{8}+\delta_{p}$	$\frac{{3\lambda }_{0}}{{4n}_{\mathrm{eff}}}+n_{\mathrm{MM}}\delta_{p}$

**Table 2 sensors-20-06804-t002:** Layer thicknesses, plasmon-enhanced absorptance, relative improvements of the GA optimization process and *Q* factors obtained for the optimized configurations.

Mirror Material	t	$d_{1}$	$d_{2}$	$d_{3}$	$d_{4}$	$d_{5}$	${a(\lambda }_{0})$	$\Delta a_{GA}$	Q
Ag	0.7	0.73	0.40	0.62	1.14	-	52.7%	1.7%	82.6
Ag	0.9	0.68	0.45	0.52	1.09	-	64.7%	4.7%	88.8
Ag	1.1	0.63	0.41	0.49	1.07	-	70.6%	5.1%	82.6
W	0.7	0.66	0.41	0.55	1.135	-	64.3%	1.7%	62.2
W	0.9	0.64	0.50	0.45	1.08	-	75.0%	2.4%	66.5
W	1.1	0.56	0.42	0.49	1.06	-	79.5%	4.3%	66.4
Mo	0.7	0.74	0.48	0.50	1.14	-	57.1%	2.8%	76.1
Mo	0.9	0.63	0.43	0.49	1.10	-	69.7%	4.2%	71.0
Mo	1.1	0.63	0.365	0.50	1.07	-	74.3%	6.2%	76.6
doped Si	0.7	0.27	0.59	0.39	0.95	0.25	65.5%	1.8%	21.5
doped Si	0.9	0.31	0.51	0.35	0.92	0.24	74.5%	11.1%	23.5
doped Si	1.1	0.30	0.49	0.39	0.89	0.23	81.5%	6.6%	23.2

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
