# Peer review of "Impact of Different Metals on the Performance of Slab Tamm Plasmon Resonators"

_sensors, 2020, doi:10.3390/s20236804_

Round 1

Reviewer 1 Report

The article presents a theoretical investigation of the performance of Slab Tamm Plasmon Resonators as a function of the metals used to fabricate them. Slab Tamm Plasmon resonators are Tamm Plasmon resonators with a novel structure that simplifies integration, which was proposed by the same group in 2019.

The article is well written and well structured. The experiment is well presented and the methodology is sound.

I just have a few comments/suggestions that can improve/clarify some of the content:

  • The figures need to be revised. Figure 1: Font of the text within the figure is unequal. The thickness is labelled by a capital W while in the text it is labelled by a cursive w. Which makes it confusing, as the authors also use tungsten below. Figures 4a, 5a, 6a - There is no way to distinguish between the lines, either use different colours or signs. I would revise all figures, colours, alignment, sizes...
  • Some definitions have to be revised. WG appears in line 368 but it is not defined. microGA appears in line 220, but is defined in line 229. There is a problem with consistency naming the layer thickness. In line 70 it appears again as capital W. a(lambda) is defined in line 234, but actually appears in line 204. Please check the whole document, because there are a number of cases. 
  • It would be good to clarify what kind of edges and surfaces of the slabs are used in the simulation. I presume they are ideal step edges? What is the tolerance and how would this affect their ability to fabricate structures while keeping the performance shown in the simulations?
  • Optical dispersions of the different materials are taken from the literature. Would the authors expect any changes due to deposition method, deposition conditions, defect density, crystallinity...?
  • I miss a clearer example of the effect of the genetic algorithm. The optimum conditions for different thicknesses found by the algorithm are summarised in table 2, but what was the performance of the resonator before the algorithm was used? Can this be included in one of the figures? The change in absorptance at resonance in table 2 provides an idea, but a figure would be much more illustrative. For example including the absorptance in figure 4a,5a,6a?
  • Table 2 results: I do not think the text makes it clear that the authors choose 3 different thicknesses and use the algorithm for those three different thicknesses for each of the metals. 
  • What is the reason for the lack of correlation between absorptance and Q factor? It is said that relative field enhancement is reflected by both a and Q. However the relative field enhancement of Silicon is low in comparison with the other metals, as is Q, but absorptance is similar to the one of the resonators made out of the other metals.
  • What is the reason behind choosing 1021cm-3 as the doping level of Silicon? Why not choosing the maximum doping that it is still possible to fabricate? and if you are to choose something that is impossible to fabricate, why not higher?
  • A few typos can be found across the document: Extra "is" in line 123, though instead of through in line 271, an instead of a in lines 292 and 294. 

Reviewer 2 Report

Reviewers comments:

Author present the concept of slab Tamm plasmon (STP) in regard of their properties as resonant absorber or emitter structures in the mid infrared wavelength. STP absorbers can mitigate the degradation of Q for lower-reflective metals or even non-metals such as doped silicon as plasmonic absorber material. I acknowledge the authors efforts in this manuscript. Si waveguide (WG) with STP studied seem very original, and therefore of interest to the community in the optical communication application. However, there are a few grammatical mistakes and ambiguities in explanation of the content. I request the authors to kindly follow the comments below before publication.

  1. In abstract, the authors mention this STP structures are compatible with modern MEMS mass fabrication processes. I don’t see any clear MEMS element in this manuscript. Can authors explain it clearly, otherwise, I suggest to remove these descriptions.
  2. Experimentally, this STP structures could be very challenging to do. How does the authors envision that such an experiment would be possible, especially the metals (Ag, W, Mo) are in the same plane of Si WG? It's better to show the fabrication process.
  3. Could authors explain how the radiation losses in the STP structures? (line 81)
  4. Could authors explain the mechanism of how the increase ohmic losses facilitate the fulfilment of the STP resonance condition? (line 145)
  5. In order to distinguish different lines in the figures, we suggest the author to use different colors rather than different thicknesses.
  6. Lines 22 (MEMS) and line 58 (TE):  Abbreviations must be used full names when they appear at first time.
  7. Why do authors use such values (0.7, 0.9, 1.1) for W? It is necessary to explain it.
  8. The authors are also suggested to check the grammar and language of the manuscript because several errors can be found along the text and the figures.
  9. For discussing the metals used in Si WG for infrared optoelectronics, the authors should reference other relative articles published earlier: 
  • Nanomaterials, vol.10, pp.1442, 2020.

Round 2

Reviewer 2 Report

This manuscript is well-organized and good quality. I recommend it can be published.